# Chemical Investigation of Endophytic *Diaporthe unshiuensis* YSP3 Reveals New Antibacterial and Cytotoxic Agents

**DOI:** 10.3390/jof9020136

**Published:** 2023-01-19

**Authors:** Babar Khan, Yu Li, Wei Wei, Guiyou Liu, Cheng Xiao, Bo He, Chen Zhang, Nasir Ahmed Rajput, Yonghao Ye, Wei Yan

**Affiliations:** 1State & Local Joint Engineering Research Center of Green Pesticide Invention and Application, College of Plant Protection, Nanjing Agricultural University, Nanjing 210095, China; 2Key Laboratory of Integrated Management of Crop Diseases and Pests, Ministry of Education, Nanjing 210095, China; 3School of Life Sciences and Chemical Engineering, Jiangsu Second Normal University, Nanjing 211200, China; 4Department of Plant Pathology, University of Agriculture, Faisalabad 38000, Pakistan

**Keywords:** antibacterial activity, cytotoxic activity, *Diaporthe unshiuensis*, endophytic fungus, xanthones

## Abstract

Chemical investigation of the plant-derived endophytic fungus *Diaporthe unshiuensis* YSP3 led to the isolation of four new compounds (**1**–**4**), including two new xanthones (phomopthane A and B, **1** and **2**), one new alternariol methyl ether derivative (**3**) and one α-pyrone derivative (phomopyrone B, **4**), together with eight known compounds (**5**–**12**). The structures of new compounds were interpreted on the basis of spectroscopic data and single-crystal X-ray diffraction analysis. All new compounds were assessed for their antimicrobial and cytotoxic potential. Compound **1** showed cytotoxic activity against HeLa and MCF-7 cells with IC_50_ values of 5.92 µM and 7.50 µM, respectively, while compound **3** has an antibacterial effect on *Bacillus subtilis* (MIC value 16 μg/mL).

## 1. Introduction

Microorganisms produce a wide range of secondary metabolites (SM), also known as natural products, which have an incredible and dignified success history concerning pharmaceutical potential and structural diversity [1]. Metabolites produced by endophytic fungal isolates are not only renowned for providing protection in the survival of their host but also have a phenomenal contribution to agriculture, medicine, and modern industry [2]. For pathogenic fungi, some natural products were used as chemical weapons to facilitate their invasion. For example, fusaoctaxin A was recently characterized as a virulence factor during the infection progress of *Fusarium graminearum*, which is a destructive wheat pathogen [3]. Moreover, for endophytic fungi, their metabolites can provide benefits to residing hosts by functioning as antibacterials, nutrition transporting agents, and plant growth regulators [3,4].

Species of *Diaporthe* (anamorph *Phomopsis*) comprise a diverse and widely distributed group of phytopathogens, saprophytes, endophytes, and pathogens of mammals [5]. As plant pathogens, they can infect a wide range of plant hosts (soybean, eggplant, grape, citrus, etc.) and cause diseases [6]. Many other species of *Diaporthe* are reported as non-pathogenic endophytes and recognized as producers of numerous secondary metabolites with a range of biological potencies, including antifungal [7], antibacterial [8], cytotoxic [9], anti-inflammatory [10], antioxidant [11], phytotoxic [12], and anti-influenza A virus (IAV) potencies [13]. Most of the compounds produced by this genus have been classified into polyketides, including xanthones, chromanones, benzofuranones, quinones, phenols and pyrones, terpenoids, steroids, lactones, alkaloids, and fatty acids [14,15].

As part of our continuous endeavor to search for structurally intriguing and/or bioactive natural products from endophytic fungi, an attempt was made to investigate the fungus *Diaporthe unshiuensis* YSP3 isolated from the leaves of *Caesalpinia sepiaria*. This work led to the isolation of two new xanthone derivatives (**1**–**2**), one new alternariol derivative (**3**), one new pyrone derivative (**4**), and eight known compounds. Herein, the isolation, structural elucidation, and bioactivity evaluation of metabolites **1**–**12** are described in detail.

## 2. Materials and Methods

### 2.1. General Experimental Procedures

Nuclear magnetic resonance (NMR) spectra were recorded on a Bruker Avance III 400 and 600 MHz NMR spectrometer (Bruker, Rheinstetten, Germany) in CDCl_3_ and acetone-*d*_6_ with TMS as an internal standard at room temperature. Ultraviolet (UV) spectra were recorded on a Hitachi U-3000 spectrophotometer (Hitachi, Tokyo, Japan). Optical rotations were measured in MeOH solution on a Rudolph Autopol III automatic polarimeter (Rudolph Research Analytical, NJ, USA). High-resolution-electrospray ionization-mass spectrometry (HR-ESI-MS) spectra were obtained on an Agilent 6210 TOF LC-MS spectrometer (Agilent, CA, USA). X-ray data were obtained on a Bruker APEX-II CCD diffractometer (Bruker, MA, USA). Column chromatography was performed on silica gel (200−300 mesh, Qingdao Marine Chemical Inc., Qingdao, China) and Sephadex LH-20 (Pharmacia Biotech, Uppsala, Sweden). High-performance liquid chromatography (HPLC) was performed on a Shimadzu LC-20AT instrument with an SPD-20A detector (Agilent, CA, USA) using an ODS column (ODS-2 HYPERSIL, 250 × 10 mm, 5 μm, Thermo Scientific, Shanghai, China). All chemicals used in this study were of analytical or HPLC grade.

### 2.2. Fungal Materials

The fungal strain YSP3 was isolated from the leaves of *C. sepiaria* collected in August 2016 from Nanjing Botanical Garden, Mem. Sun Yat-Sen, Nanjing, Jiangsu Province, People’s Republic of China. The strain was identified as *D. unshiuensis* (GenBank OP804247) based on a morphological characterization as well as phylogenetic analysis using five molecular markers (ITS, TEF1, HIS, CAL, and TUB) (Appendix A) [16]. The strain YSP3 was deposited in the culture collection bank of the Laboratory of Natural Products and Pesticide Chemistry, Nanjing Agricultural University (NAU).

### 2.3. Fermentation, Extraction, and Purification

Strain YSP3 was cultured on potato dextrose agar medium (PDA) for 5 days at 25 °C. Seed medium (potato 200 g, dextrose 20 g, distilled water 1000 mL) in 1000 mL Erlenmeyer flasks containing 400 mL broth was inoculated with pieces of the mycelium obtained from the colony strain YSP3 and incubated for 48 h at 25 °C on a rotating shaker (150 rpm). Fermentation was performed in 150 Erlenmeyer flasks containing PD medium (400 mL in each) inoculated with 10 mL seed solution and incubated at 25 °C for 14 days on a rotary shaker (150 rpm). The fermented broth (60 L) was filtered through muslin cloth, extracted thrice by adding EtOAc (*v*/*v*), and evaporated under vacuum to obtain crude extract (41 g). The EtOAc extract was separated into six fractions (Fr1-Fr6) on the silica gel column (350 g, 200−300 mesh), eluted with a gradient solvent system of CH_2_Cl_2_-MeOH (*v*/*v* 100:0, 100:1, 100:2, 100:4, 100:8, 100:16, 0:100). Fr2 was subjected to the silica gel (300−400 mesh), eluted with a gradient of petroleum ether–EtOAc (*v*/*v*, 100:0 0:100) to give five subfractions (Fr2.1 to Fr2.5). Fr2.2 was chromatographed over a Sephadex LH-20 (MeOH) column and then separated by semipreparative HPLC (MeOH/H_2_O, *v*/*v*, 50:50, 0.05% trifluoroacetic acid (TFA)) to obtain compound **2** (10 mg, *t*_R_ = 35.7 min). Fr2.3 was further purified by Sephadex LH-20 (MeOH) and semipreparative HPLC using 42% (MeOH) in water to obtain **3** (7.2 mg, *t*_R_ = 16.3 min) and **4** (38 mg, *t*_R_ = 32 min). Fr3 was separated by silica gel chromatography (petroleum ether–EtOAc *v*/*v*, 100:0 to 0:100) to give five subfractions (Fr3.1 to Fr3.5). Fr3.2 was separated by HPLC (54% MeOH in H_2_O, flow rate = 2 mL/min, UV = 254 nm) to yield **5** (6.7 mg, *t*_R_ = 13.2 min), **6** (5.9 mg, *t*_R_ = 30.7 min), and **9** (8.7 mg, *t*_R_ = 24.2 min). Fr3.3 was subjected to the Sephadex LH-20 (MeOH) chromatography and further purified by HPLC (MeOH/H_2_O, *v*/*v*, 45:55, 0.05 % TFA) to yield **1** (12.4 mg, *t*_R_ = 19.3 min), **7** (6.2 mg, *t*_R_ = 21 min), and **8** (7 mg, *t*_R_ = 16.9 min). Fr3.4 was further separated by HPLC eluted with a gradient of (MeOH/H_2_O, *v*/*v*, 60:40) to afford compound **11** (7.8 mg, *t*_R_ = 26 min). Fr4 was divided into four subparts (Fr4.1 to Fr4.4) by silica gel column with a gradient elution of petroleum ether–EtOAc (*v*/*v*, 100:0–0:100). Fr4.2 was subjected to a Sephadex LH-20 (MeOH) column and further purified by semipreparative HPLC (MeOH/H_2_O, *v*/*v*, 65:35) to afford **10** (7.2 mg, *t*_R_ = 35.7 min) and **12** (12 mg, *t*_R_ = 49 min).

### 2.4. Formation of the (S)- and (R)-MTPA Esters of ***4***

Compound **4** (2.0 mg) was dissolved in 500 μL of anhydrous pyridine, and 20 μL of (*S*)-(+)-MTPA chloride was added to the reaction mixture. After 48 h reaction in dark, the residue was purified by HPLC (CH_3_CN/H_2_O, 85:15) to give (*R*)-(+)-MTPA ester (1.7 mg). (*R*)-(−)-MTPA ester (1.8 mg) was obtained using the same step from (*S*)-(−)-MTPA chloride.

### 2.5. Phomopthane A (***1***)

Colorless crystals; [α]^20^_D_ -96.0 (c 0.1, MeOH); UV (MeOH) *λ*_max_ (log *ε*) 210 (1.9), 280 (1.0), 360 (0.4) nm; ^1^H and ^13^C NMR data, see Table 1; HR-ESI-MS *m*/*z* 331.0789 [M + Na]^+^ (cacld for C_15_H_18_O_3_Na, 331.0788).

### 2.6. Phomopthane B (***2***)

Colorless crystals; [α]^20^_D_ +4.0 (c 0.1, MeOH); UV (MeOH) *λ*_max_ (log *ε*) 210 (1.8), 280 (1.0), 360 (0.4) nm; ^1^H and ^13^C NMR data, see Table 1; HR-ESI-MS *m*/*z* 333.0950 [M + Na]^+^ (cacld for C_15_H_18_O_4_Na, 333.0945).

### 2.7. Alternariol Methyl Ether-12-O-α-D-arabinoside (***3***)

Yellow powder; [α]^20^_D_ +77.9 (c 0.1, MeOH); UV (MeOH) λ_max_ (log *ε*) 589 nm; ^1^H and ^13^C NMR spectroscopic data, see Table 2; HR-ESI-MS *m*/*z* 427.1006 [M + Na]^+^ (cacld for C_20_H_20_O_9_Na, 427.1005).

### 2.8. Phomopyrone B (***4***)

Pale yellow oil; [α]^20^_D_ -41.4 (c 0.2, MeOH); UV (MeOH) λmax (log *ε*) 589 nm; ^1^H and ^13^C NMR spectroscopic data, see Table 2; HR-ESI-MS *m*/*z* 213.1129 [M + H]^+^ (cacld for C_11_H_17_O_4_, 213.1127).

### 2.9. X-ray Crystallographic Analysis

The single colorless crystals of compounds **1** and **2** were shaped from the MeOH + CH_2_Cl_2_ mixture solvent (*v*/*v*, 1:1/2) at 4 °C after 25 or 27 days of slow solvent evaporation. Crystal diffraction data were collected on a Bruker APEX-II CCD diffractometer using Cu Kα radiation (λ = 1.5418 Å). These structures were solved by direct methods in OLEX2–1.3 software, followed by the refinement method of full-matrix least-squares calculations on F2 using SHELXL-2018 [17,18]. All crystal data of these compounds have been deposited in the Cambridge Crystallographic Data Centre (CCDC).

### 2.10. Crystal Data of ***1***

Triclinic, space group *P*-1, a = 8.2397 (3) Å, b = 8.5629 (3) Å, c = 11.4399 (4) Å, α = 69.676 (2)°, β = 86.302 (2)°, γ = 75.660 (2)°, V = 733.11 (5) Å3, Z = 2, T = 296 (2) K, Dcalc = 1.478 g/cm^3^, μ(Cu Kα) = 1.035 mm^−1^, F (000) = 344.0, Crystal size = 0.12 × 0.11 × 0.09 mm^3^, 4349 reflections collected, 2407 independent reflections (Rint = 0.0183). Final R1 = 0.0355, wR2 = 0.1007 [I >= 2σ(I)]. Final R1 = 0.0374, wR2 = 0.1024 (all data). The goodness of fit on F2 was 1.071. CCDC number: 2083291.

### 2.11. Crystal Data of ***2***

Monoclinic, space group *P*2_1_/*c*, a = 11.3791 (11) Å, b = 7.4153 (6) Å, c = 16.9366 (15) Å, α = 90°, β = 108.048 (5)°, γ = 90°, V = 1358.8 (2) Å3, Z = 4, T = 296 (2) K, Dcalc = 1.517 g/cm^3^, μ(Cu Kα) = 1.027 mm^−1^, F (000) = 658.6, Crystal size = 0. 12 × 0.11 × 0.09 mm^3^, 5641 reflections collected, 2426 independent reflections (Rint = 0.0213). Final R1 = 0.0374, wR2 = 0.1092 [I >= 2σ(I)]. Final R1 = 0.0393, wR2 = 0.1118 (all data). The goodness of fit on F2 was 1.074. CCDC number: 2083288.

### 2.12. Antimicrobial Assays

All isolated new compounds were evaluated for their antibacterial potencies against four bacterial strains (*Xanthomonas oryzae* pv. *oryzae*, *Xanthomonas oryzae* pv. *oryzicola*, *Bacillus subtilis,* and *Ralstonia solanacearum*) in sterile 96-well plates by a broth dilution method [19]. Antifungal potencies were performed against *Rhizoctonia solani*, *Fusarium solani*, *F. graminearum*, *Botrytis cinerea*, *Sclerotinia sclerotiorum*, *Alternaria solani,* and *Phytophthora capsici* referred to the method reported [20]. All strains were provided by the Laboratory of Natural Products and Pesticide Chemistry, Nanjing Agricultural University, Nanjing, Jiangsu, China.

### 2.13. Cytotoxic Assays

The inhibitory effects of isolated compounds on the HeLa and MCF-7 cells were assessed in vitro by using the MTT assay on a 96-well plate. Experimental details of test used in this study are summarized in previous reports [21,22]. The toxicities against normal LO2 and HaCaT cells were tested via the cell counting kit-8 (CCK-8) method, similar to the reported reference [23].

### 2.14. Statistical Analysis

All tests were repeated thrice to remove the experimental error, and the quantitative data were presented as mean values ± standard deviation. The data were analyzed using IBM SPSS 22.0 with the probit analysis, and the Duncan statistical test was used for variance analysis between means. The value *p* ≤ 0.05 was considered statistically significant.

## 3. Results and Discussion

Phomopthane A (**1**) was obtained as colorless crystals with the molecular formula C_15_H_16_O_7_ as inferred from its HRESIMS, (*m*/*z* 331.0789 [M + Na]^+^, cacld for C_15_H_16_O_7_Na, 331.0788), indicating eight degrees of unsaturation. The ^13^C and DEPT NMR spectra exhibited 15 resonances resulting from one methyl, two methylenes, five methines, and seven quaternary carbons, including two carbonyl groups (*δ*_C_ 204.5 and *δ*_C_ 196.9). The ^1^H NMR spectrum revealed the presence of characteristic signals for three aromatic protons from a 1,2,3- trisubstituted benzene at *δ*_H_ 6.54 (d, *J* = 8.2 Hz, H-2), 6.48 (d, *J* = 8.2 Hz, H-4), and 7.47 (t, *J* = 8.2 Hz, H-3), one methylene at 2.14 (m, H-9), one oxygenated methylene at *δ*_H_ 4.83 (d, *J* = 13.1 Hz, H-14a) and 3.96 (d, *J* = 13.1 Hz, H-14b), two methines at *δ*_H_ 4.61 (br t, *J* = 2.5 Hz, H-10) and 3.12 (m, H-8), and one methyl group at *δ*_H_ 1.08 (d, *J* = 6.4 Hz, H-15). The ^1^H and ^13^C NMR spectra of **1** (Table 1) were closely similar to those of mangrovamide K [24], a xanthone derivative isolated from *Penicillium* sp.. The difference was attributed to the methyl group in mangrovamide K being changed to the oxygenated methylene at C-14 in structure **1** (Figure 1), which was further supported by the HMBC correlations from H_3_-15 to C-7, C-8, and C-9; H-14 to C-6 and C-7; and H-9 to C-7, C-11, and C-10 (Figure 2). Thus, the planar structure of **1** was proposed, as shown in Figure 1.

The relative configuration of **1** was partially determined by NOESY and 3JHH coupling data. H-10 showed a small coupling constant (2.5 Hz) to H-9, suggesting H-10 has an equatorial orientation. The NOESY correlation between H-8 and H-14 indicates that they both possess an axial position (Figure 3). There was no sufficient data to deduce the configuration at C-11. Fortunately, a single crystal X-ray study not only confirmed the planar structure but also determined the relative configuration of **1** (Figure 4).

The absolute configuration of **1** was assigned by comparing its electronic circular dichroism (ECD) spectrum with the structurally similar mangrovamide K, of which the absolute configuration has been established. Compound **1** contains the same chromophoric system as mangrovamide K but shows nearly a mirror image on ECD (Figure 5) [24], with a strong positive Cotton effect at 211 nm, negative bands in the 250–350 nm range, and a weak positive Cotton effect at 359 nm. Additionally, **1** has an opposite optical rotation ([α]^20^_D_ −96.0) compared to mangrovamide K ([α]^25^_D_ +72.9). Therefore, the absolute configuration of **1** was determined as *6R*,*8R*,*10S*,*11S*.

Phomopthane B (**2**) was isolated as colorless crystals. Its molecular formula, C_15_H_18_O_7_, was deduced by HR-ESI-MS (*m*/*z* 333.0950, [M + Na]^+^, calcd for C_15_H_18_O_7_Na, 333.0945), which was two more mass units than **1**. The close comparison of the 1D NMR data between **2** and **1** showed a general similarity (Table 1), except that the ketone at the C-7 position in **1** was replaced by an oxygenated methine at *δ*_H_ 4.42 (H-7) in **2**. This assumption was supported by the key HMBC correlations of H_3_-15 to C-7, C-8, and C-9; H-14 to C-6 and C-7; and H-9 to C-7 and C-11 (Figure 2). H-7 was suggested to have an equatorial position, evidenced by its small *J* value (2.8 Hz). The relative configuration of the rest chiral centers in **2** was the same as in **1**, based on their similar NOESY correlations and coupling constants (Figure 3 and Table 1). The above-mentioned structural elucidation was confirmed by a single X-ray analysis. Unlike **1**, **2** crystallized in the monoclinic *P*2_1_/*c* space group and has a near-zero optical rotation, suggesting that **2** was a racemic mixture [25,26].

Compound **3** was obtained as a gray powder with a molecular formula of C_20_H_20_O_9_ determined by the HR-ESI-MS spectrum (*m*/*z* 427.1006 [M + Na]^+^; calcd for C_20_H_20_O_9_Na, 427.1000), indicating 11 degrees of unsaturation. The ^13^C NMR, DEPT, and HSQC spectra revealed the presence of 20 carbons signals resulting from for one methoxy, one methyl, one methylene, eight methines, and nine quaternary carbons, including one lactonic ester group at *δ*_C_ 165.7 (C-9). The ^1^H-NMR data (Table 2) displayed characteristic resonances for four aromatic protons at *δ*_H_ 6.61 (d, *J* = 2.2 Hz, H-1), 7.00 (2H, m, H-11, and H-13), and 7.34 (d, *J* = 1.9 Hz, H-3), one methoxy group at *δ*_H_ 3.99 (s, H-15), one methyl group at *δ*_H_ 2.85 (s, H-14), one oxygenated methylene at *δ*_H_ 3.71 (m, H-5′), four oxygenated methines at *δ*_H_ 4.20 (2H), 4.31, and 5.80. The HMBC correlations (Figure 2) from H_3_-15 to C-2 and C-6; H_3_-14 to C-10, C-7, and C-11; H-13 to C-8 and C-11 suggested the presence of alternariol methyl ether in the chemical structure of **3 [27]**. The NMR data (Table 2) of compound **3** were very similar to those of **8**, except the existence of five more oxygenated carbons at *δ*_C_ 101.5 (C-1′), 73.1 (C-2′), 88.2 (C-3′), 70.8 (C-4′), and 62.9 (C-5′) in **3**. Considering the molecular formula, it is speculated that there should be a monosaccharide structure in **3**, which was further confirmed by the COSY correlations of H-1′/H-2′/H-3′/H-4′/H-5′ and HMBC correlations from H-1′ to C-2′, C-3′, and C-4′. The HMBC correlation of the anomeric proton at *δ*_H_ 5.80 (d, *J* = 4.3 Hz) with C-12 (*δ*_C_ 158.7) determined its linkage site in compound **3**. The chemical shift of the anomeric carbon (*δ*_H_ 5.80 and *δ*_C_ 101.5) confirmed the α configuration. The sugar moiety was further determined as D-arabinoside by hydrolysis and compared to the standard. Thus, **3** was named as alternariol methyl ether-12-*O*-α-D-arabinoside shown in (Figure 1).

Compound **4** was isolated as a pale yellow oil. Its molecular formula, C_11_H_16_O_4_, was deduced by HR-ESI-MS (*m*/*z* 213.1129, [M + H]^+^, calcd for C_11_H_17_O_4_, 213.1121), indicating four degrees of unsaturation. The ^13^C and DEPT NMR data demonstrated 11 resonances resulting from two methyls, one methoxy, two methylenes, two methines, and four quaternary carbons signals, including an ester carbonyl group at (*δ*_C_ 164.9, C-2). The ^1^H-NMR spectral data (Table 2) revealed the signals for one olefinic proton at *δ*_H_ 7.48 (s, H-6), one methoxy group *δ*_H_ 3.95 (s, H-12), two methyl groups at *δ*_H_ 1.99 (s, H-11) and *δ*_H_ 0.92 (t, *J* = 7.4 Hz, H-10), two methylenes at *δ*_H_ 1.50 (m, H-9) and at *δ*_H_ 1.70 (m, H-8), and one methine at *δ*_H_ 4.63 (m, H-7). The α-pyrone ring structure was confirmed by HMBC correlations (Figure 2) from H-6 to C-2, C-4, and C-5. HMBC correlations from H-7 to C-8, C-9, and COSY correlations of H-7/H-8/H-9/H-10 determined the presence of a butyl side chain. The attachment of the butyl side chain at C-5 was confirmed by HMBC correlation from H-7 to C-4, C-5, and C-6. The attachment of the methoxy group (C-12) was also confirmed by the HMBC correlation between H-12 and C-4. Thus, the planar structure of **4** was deduced, as shown in Figure 1.

The absolute configuration of C-7 in **4** was confirmed by a modified mosher’s method [28,29]. Compound **4** reacted with (*R*)-(−)-MTPA chloride and (*S*)-(+)-MTPA chloride, respectively, to give the (*S*)- and (*R*)-MTPA esters. The Δ*δ* values between *R*- and *S*-MTPA derivatives confirmed that C-7 was an *R* configuration (Figure 6). Finally, compound **4** was named phomopyrone B.

In addition, the other known compounds (**5**–**12**) were identified as 6,8-dihydroxy-3-methyl-9-oxo-9H-xanthene-1-carboxylic acid (**5**) [30], 3,8-dihydroxy-6-methyl-9-oxo-9H-xanthene-1-carboxylic acid (**6**) [31], alternariol (**7**), alternariol methyl ether (**8**) [32], monodictyphenone (**9**) [33], 3,4-dihydro-6,8-dihydroxy-3-methylisocoumarin (**10**) [34], 2-(2′S-hydroxypropyl)-7-hydroxychromone’s (**11**) [35], and wermopyrone (**12**) [36], respectively, by comparing their HRMS and NMR data to the literature.

To fully evaluate the antimicrobial potencies, newly isolated compounds (**1**–**4**) were screened for their antibacterial effect against four bacterial strains (three Gram-negative phytopathogenic bacteria and one Gram-positive bacterium) as well as for antifungal activities against seven fungal strains, respectively. Compound **3** showed the bactericidal effect against *B. subtilis* with an MIC value of 16 μg/mL, which was better than the positive control (streptomycin sulfate 64 μg/mL) (Table 3). The compounds (**1**–**4**) were also tested for their cytotoxic activities by the MTT test method. Compound **1** was active against HeLa and MCF-7 cell lines with IC_50_ values of 5.92 ± 0.04 µM and 7.50 ± 0.02 µM, respectively. Colchicine was used as a positive control (0.36 ± 0.07 µM for Hela and 0.44 ± 0.18 µM for MCF-7 cell lines) (Table 4). The other compounds were found to be ineffective (IC_50_ values > 20 µM). To explore the side effects of compound **1** on normal cells, **1** was tested for the toxicities towards normal LO2 and HaCaT cells using the CCK-8 assay. As shown in Appendix A, compound **1** showed little effect on the viability of both LO2 and HaCat cells, which maintained high cell viability (~90 %) even at the compound concentration of 100 μM. These results indicated that **1** has a selective cytotoxic effect on tumor cell lines.

## 4. Conclusions

The current investigation reported the isolation and structural elucidation of twelve secondary metabolites, including two new xanthone derivatives (**1** and **2**), one new alternariol derivative (**3**), one new pyrone derivative (**4**), along with eight known compounds (**5**–**12**). All new compounds were evaluated for their bioactivities (antifungal, antibacterial, and cytotoxic). Compound **1** revealed potent cytotoxic potencies against human cancer cell lines HeLa and MCF-7, while compound **3** showed a bactericidal effect on *B. subtilis.* Additionally, **1** exhibited low toxicity to normal cells (LO2 and HaCaT). Thus, taking the importance of natural products, these findings enriched the structural diversity of secondary metabolites from *Diaporthe* species and highlighted their values for pharmaceutical or bactericidal applications.

## Figures and Tables

**Figure 1 jof-09-00136-f001:**
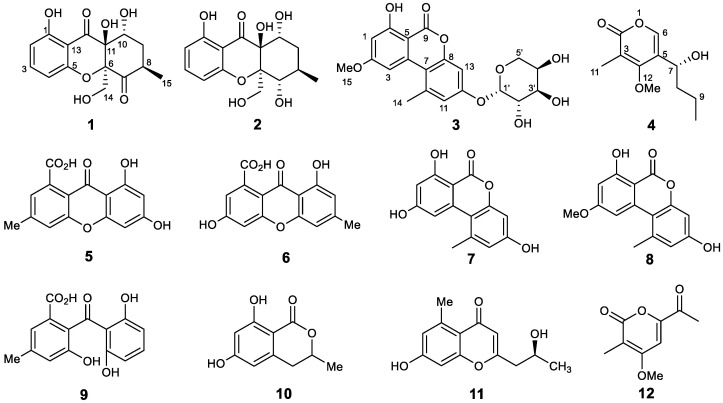
Structures of isolated compounds (**1**–**12**).

**Figure 2 jof-09-00136-f002:**
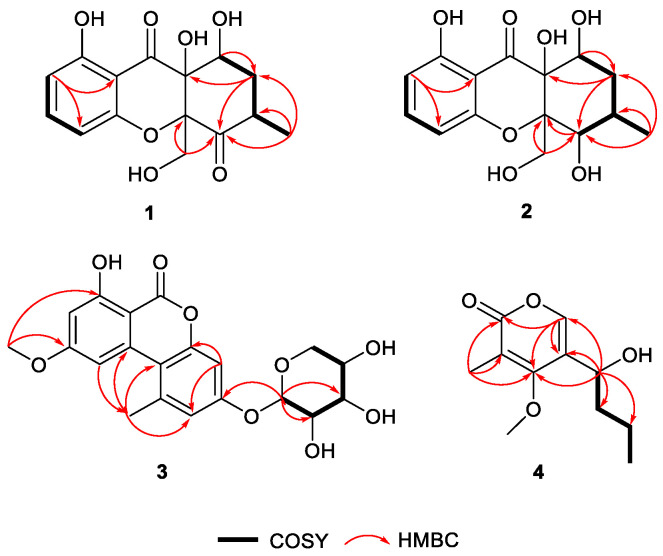
Key ^1^H-^1^H COSY and HMBC correlations of compounds **1**–**4**.

**Figure 3 jof-09-00136-f003:**
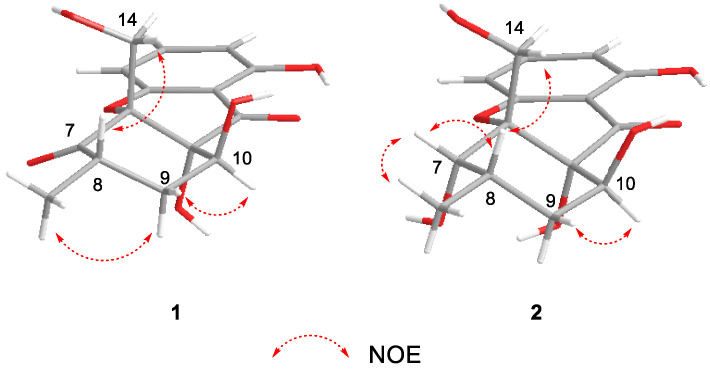
Key NOESY correlations of **1** and **2**.

**Figure 4 jof-09-00136-f004:**
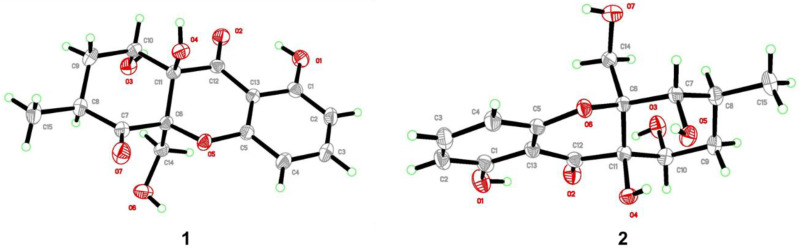
X-ray crystallographic structures of **1** and **2**.

**Figure 5 jof-09-00136-f005:**
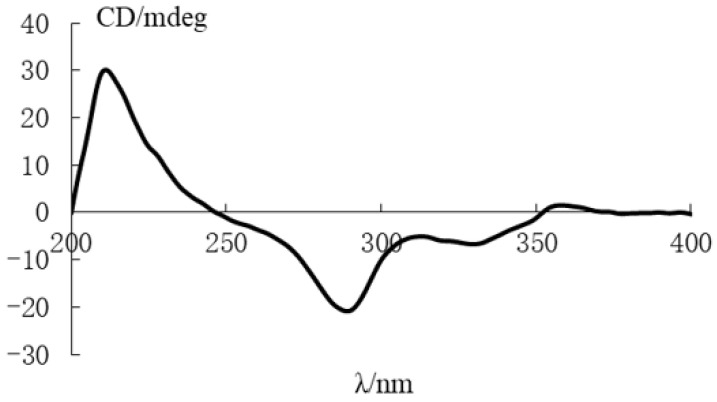
The ECD spectrum of **1**.

**Figure 6 jof-09-00136-f006:**
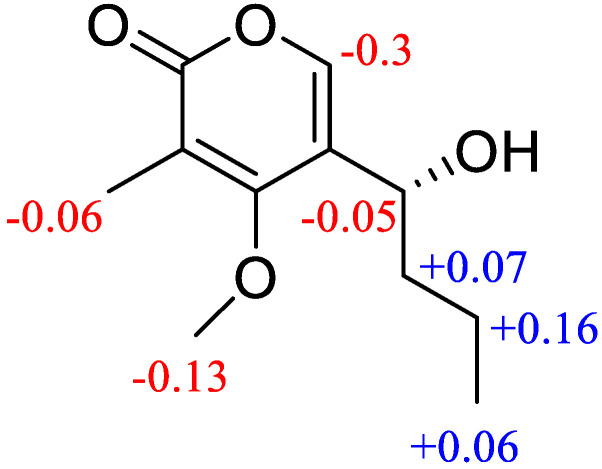
Δ*δ* values (*δ_S_* − *δ_R_*) obtained for the *S*- and *R*-MTPA esters of compound **4**.

**Table 1 jof-09-00136-t001:** ^1^H NMR (600 MHz) and ^13^C NMR (150 MHz) spectral data of **1** and **2**.

	1 (Acetone*-d_6_*)		2 (CDCl_3_)	
No.	*δ*_C,_ Type	*δ*_H_ (*J* in Hz)	*δ*_C,_ Type	*δ*_H_ (*J* in Hz)
1	163.8, C		160.0, C	
2	110.6, CH	6.54 (d, 8.2)	105.3, CH	6.54 (d, 8.1)
3	140.2, CH	7.47 (t, 8.2)	136.1, CH	7.42 (t, 8.1)
4	110.5, C	6.48 (d, 8.2)	108.0, CH	6.58 (d, 8.1)
5	160.0, C		155.2, C	
6	94.3, C		81.8, C	
7	204.5, C		72.2, CH	4.42 (d, 2.1)
8	38.3, CH	3.12 (m)	25.3, CH	2.35 (m)
9	39.5, CH_2_	2.14 (m)	27.1, CH_2_	1.73 (m)
				2.09 (m)
10	67.8, CH	4.61 (br t, 2.5)	65.8, CH	4.53 (s)
11	79.6, C		71.9, C	
12	196.9, C		193.2, C	
13	108.4, C		105.0, C	
14	63.5, CH_2_	4.83 (d, 13.1)	58.1, CH_2_	3.88 (d, 13.5)
		3.96 (d, 13.1)		4.20 (d, 13.5)
15	15.4, CH_3_	1.08 (d, 6.4)	15.1, CH_3_	1.17 (d, 6.7)
1-OH				10.8 (s)

**Table 2 jof-09-00136-t002:** ^1^H NMR (600 MHz) and ^13^C NMR (150 MHz) spectral data of **3** and **4** in acetone-*d*_6_.

	3		4	
No.	*δ*_C_, Type	*δ*_H_ (*J* in Hz)	*δ*_C_, Type	*δ*_H_ (*J* in Hz)
1	100.3, CH	6.61 (d, 2.2)		
2	167.5, C		164.9, C	
3	105.1, CH	7.34 (d, 1.9)	110.5, C	
4	138.7, C		166.4, C	
5	99.9, C		121.7, C	
6	165.8, C		146.4, CH	7.48 (s)
7	112.2, C		66.0, CH	4.63 (m)
8	153.7, C		39.4, CH_2_	1.70 (m)
9	165.7, C		18.7, CH_2_	1.50 (m)
10	139.3, C		13.2, CH_3_	0.92 (t, 7.4)
11	119.5, CH	7.00 (m)	9.8, CH_3_	1.99 (s)
12	158.7, C		60.7, CH_3_	3.95 (s)
13	103.8, CH	7.00 (m)		
14	25.7, CH_3_	2.85 (s)		
15	56.3, CH_3_	3.99 (s)		
1’	101.5, CH	5.80 (d, 4.3)		
2’	73.1, CH	4.31 (t, 5.2)		
3’	88.2, CH	4.20 (m)		
4’	70.8, CH	4.20 (m)		
5’	62.9, CH_2_	3.71 (m)		

**Table 3 jof-09-00136-t003:** Antibacterial potencies of compounds (**1**–**4**) against *Bacillus subtilis*.

Compounds	MIC (μg/mL)
	*Bacillus subtilis*
**1**	>100
**2**	>100
**3**	16
**4**	>100
Streptomycin sulfate	64

**Table 4 jof-09-00136-t004:** Cytotoxic effects of new compounds (**1**–**4**) on human cancer cell lines.

Compounds	Growth Inhibition IC_50_ (µg/mL) Values	
	Hela	MCF-7
**1**	5.92 ± 0.04	7.50 ± 0.02
**2**	>20	>20
**3**	>20	>20
**4**	>20	>20
Colchicine	0.36 ± 0.07	0.44 ± 0.18

## Data Availability

The data presented in the manuscript are available on request from the corresponding authors.

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
