# Peer review of "Chemical Investigation of Endophytic Diaporthe unshiuensis YSP3 Reveals New Antibacterial and Cytotoxic Agents"

_jof, 2023, doi:10.3390/jof9020136_

Round 1
Reviewer 1 Report
The paper is interesting, quite well witten and organized; interesting are the data reported, which look properly analyzed. Some notes are on the attached file

Reviewer 2 Report
The manuscript "Chemical investigation of endophytic Diaporthe unshiuensis YSP3 reveals new antibacterial and cytotoxic agents" (JofF-2103647) describes the isolation of twelve compounds from an endophytic fungus: four novel compounds and eight yet known. The work is well done and interesting, but some concerns should be addressed before publication.
An important problem is that the authors use two tumor lines to check the cytotoxicity of the compounds, but they do not include any normal cell line in the assay so that it can be known if there is a difference between this line and the two tumor lines. I believe that this data is necessary to have a complete picture of the possible usefulness of the new compounds.
Another point is that the antimicrobial activity is contrasted against three species of bacteria (the fourth is a variety), without justifying the choice of these. Obviously, all of them are phytopathogenic bacteria, but what is relevant for a possible widespread use is that Bacillus subtilis is Gram-positive and the other three are Gram-negative. And that is not mentioned anywhere...
Other aspects that may help to improve the manuscript are listed below:
- Line 23. Keywords are usually ordered in an alphabetical manner.
- Line 39. “phytopathgens” must be changed to phytopathogens.
- Lines 39-40. “pathogens of other mammals”. Please, delete “other”.
- Line 44. “anti-IAV”. IAV has not been described before.
- Line 47. “alkaloids, and fatty acid”. It is plural, fatty acids.
- Line 71. “5μm”. Please, separate amount and unit, as has been done in the rest of the document.
- Line 85. “with pieces of the mycelia“. It is singular: mycelium.
- Line 249. Please, separate Table 2 and the paragraph beginning at line 249.
- Line 284. “posi-tive control “. Please, change to “positive control”.
- Table 3. MIC (μg/ml). Please change the unit for liter to L as has been done in the rest of the document.
